# Determining the Influencing Factors of Biogas Technology Adoption Intention in Pakistan: The Moderating Role of Social Media

**DOI:** 10.3390/ijerph17072311

**Published:** 2020-03-30

**Authors:** Zanxin Wang, Saqib Ali, Ahsan Akbar, Farhan Rasool

**Affiliations:** 1School of Business and Tourism Management, and School of Development Studies, Yunnan University, Kunming 650091, China; wzxkm@hotmail.com; 2Department of Management Sciences, COMSATS University Islamabad (Sahiwal Campus), Sahiwal 57000, Pakistan; farhanrasool9966@gmail.com; 3International Business School, Guangzhou College of South China University of Technology, Guangzhou 510080, China; akbar@gcu.edu.cn

**Keywords:** renewable energy, biogas technology, norm activation model, social media, Pakistan

## Abstract

Environmental degradation and rapid climate change have forced researchers and practitioners to find sustainable practices to save the world. Increasing energy demand is not only consuming scarce natural resources, but also damaging the climate and overall ecosystem. In this regard, biogas technology is beneficial in two ways—by meeting the energy demand and saving natural resources. Pakistan is an agricultural country and has a high potential for producing energy through biogas technology. Therefore, this study aims to find farmers’ intentions of adopting biogas technology in Pakistan by employing the extended norm activation model. Furthermore, the moderating role of social media was explored. Purposive sampling was used to collect data from farmers and results were extracted by using Partial least square structural equation modelling software. The results suggest that awareness of consequences, ascription of responsibility, environmental concern and perceived consumer effectiveness positively and significantly influence personal norms of the farmers. Consequently, personal norms affect farmers’ intentions of adopting biogas technology in Pakistan. The moderating role of social media was also confirmed by the results. This study considers the notable insights of biogas technology adoption in Pakistan. Finally, the limitations of the study and suggestions for future research are discussed.

## 1. Introduction

The concept of environmental sustainability and protection has become a major topic in research and policy agendas in the past few decades because climate change is the most critical factor that causes adverse impacts on the sustainable development of the global economy [1]. Environmental deterioration, the increasing demand for energy, and the scarceness of nonrenewable sources of energy have forced countries to shift from the conventional sources of energy to renewable energy sources worldwide [2]. Renewable energy techniques address environmental issues and provide sustainable solutions for such energy problems [3]. Due to fast-paced industrialization, the demand for energy is increasing, and this increase is not only resulting in higher prices, but also improved economic development and living standards [4]. The world population is growing significantly, and lifestyles are also improving, leading to a higher demand for energy all over the world [5]. A gap between the demand and supply of energy creates hurdles in economic growth, development, prosperity and sustainable development, which have adverse effects on water resources, the environment, human health, and agricultural productivity in a country [6,7].

Most of the energy is produced with fossil fuels worldwide, but energy demand and consumption are increasing with every passing day. Fossil fuel resources are not enough to meet the desired needs, and their unstable prices have a negative impact on the world’s economy [8]. In the global south, the population is more than 650 million, from which 250 million people are still using traditional sources of energy for heating, cooking, lighting, and other daily needs [9]. Increased dependence or overuse of fossil fuels and fuel wood severely damages the ozone layer and produces an amplified degree of contamination with carbon emission, sulfur dioxide, and other harmful gases. Even though fossil fuels have adverse effects on the environment, the world is still using such nonrenewable power sources on a large scale. There is a dire need to explore and adopt renewable energy resources to cope up with the energy crisis, as well as to overcome environmental issues. Recently, most countries have explored eco-friendly and sustainable sources of energy, such as solar systems, wind energy and biogas technology [10].

Biogas is considered the most probable and potential source to address these energy and environmental issues, especially in developing countries like Pakistan. Biogas is produced through the anaerobic digestion process in which animals’ manure is mixed with water and is converted to methane gas in an airtight underground tank [11]. Biogas can not only put an end to the energy shortfall, but additional energy can be offered in the upcoming years. Biogas provides several benefits in terms of electricity generation, gas generation, bio-fertilizer, socioeconomic uplift and environmental preservation because biogas does not produce air contaminants like carbon dioxide, sulfur oxide, nitrogen oxide and other harmful fumes [12].

Pakistan is the sixth-largest country in the world in terms of population, with an annual population growth of 2%, and has 213 million residents [13]. With this growth and economic development, residents are facing multifaceted energy problems in Pakistan, due to which Pakistan experienced an energy shortage due to its high dependence on fossil fuels for power generation in recent years. During the period of 2011–2013, Pakistan has faced severe energy crises and load shedding problems [14]. Basically, Pakistan relies on conventional thermal resources for power generation, and these energy resources are not only expensive, but also unsustainable and harmful to the environment. Hence, there is a dire need to explore renewable energy alternatives in Pakistan.

Agriculture has an essential role in supporting economic growth and sustainability all around the globe. Pakistan is an agricultural economy; agricultutre plays a significant role in employment generation and contributes almost 22 percent of the gross domestic Product (GDP) of Pakistan. A survey conducted in 2012–2013 highlighted that the livestock sector contributed 56 percent of the agriculture in Pakistan, which is approximately 12 percent of GDP in the country [11]. Almost 95 percent of rural families in Pakistan use conventional biofuel sources to meet their energy and cooking needs. They use livestock manure, woody (forest, farm, and urban trees) and non-woody (crop shells, straws, leaves) materials [15]. The use of fuel wood on a large scale creates environmental hazards such as soil erosion, deforestation, flooding, land degradation and land dryness. Moreover, it has adverse effects on human health due to the accumulation of smoke in poorly ventilated houses [16].

Biogas technology has the potential to produce 8.8–17.2 billion cubic meters of gas per year from animal dung and other residues. Moreover, the bagasse, which is a residue of sugarcane and sorghum processes, has the potential to produce 5700 GWh electricity annually, and this is almost equal to 6.6% of existing power generation in Pakistan [17].

Rural farmers can gain multiple advantages from these biogas energy resources because these biogas plants can generate electricity, gas generation and bio-fertilizer, and this is also an eco-friendly technology. Pakistan has the great potential to produce almost 21 million tons of bio-fertilizer every year among gas and electricity generation [18]. The slurry, which is produced during the power generation process of biogas, is used as a bio-fertilizer. The slurry is the fibrous material or inorganic entities that cannot be broken down nor converted into methane gas, but it contains numerous fruitful and beneficial elements such as phosphorus, nitrogen, iron, cadmium, zinc, chromium, sodium, calcium and potassium [19]. Livestock manure can minimize the cost of organic fertilizers and also improve the productivity of the soil by acting as a renewable energy source. Biogas technology offers multiple economic, social and environmental benefits, but its adoption rate is very slow in developing countries such as Pakistan, especially for small farmers. Pakistan has the capacity to install around 15 million biogas plants in the country due to the abundance of this particular source [13]. In Pakistan, household biogas technology was introduced in 1959. However, there are still a limited number of biogas plants in the country (approximately 5357), even though the country has an enormous potential to install many more plants [20]. In spite of social and environmental benefits, the adoption rate of biogas technology in Pakistan is extremely low [2,3]. Hence, at this juncture, it is valuable to understand potential factors that trigger biogas technology adoption [2,20] particularly in the context of a developing country [3] such as Pakistan.

The previous literature indicates that altruistic values (concern for others) and egoistic values (pro-self) are the two main domains to predict pro-environmental behavior [21]. The literature identified that altruistic values and egoistic values are negatively associated with each other [22]. To think about inner satisfaction or for others are two different things, and both of these visions are practically incompatible and mismatched. On the other hand, both of these distinct values may exist in a person and may influence his or her attitude [23]. Egoistic values like installation cost, performance or outcomes and altruistic values like environmental concerns influence farmers’ acceptance of biogas technology. Meanwhile, the nature of biogas adoption constitutes both egoistic and altruistic values. However, extant studies mainly focused on egoistic values for biogas adoption, e.g., [3,20,24], but the role of altruism (pro-environmental) on biogas adoption was ignored or not thoroughly investigated.

To overcome this shortcoming, we study biogas adoption from an altruistic (pro-environmental) perspective, based on the norm activation model. The norm activation model (NAM) is one of the most exquisite descriptive models to measure altruistic behavior. Essentially, NAM postulates that pro-environmental behavior is predicted by personal norms [25]. Researchers in pro-environmental behavior agreed that personal norms are activated by four factors: aspiration of responsibility (AR), awareness of consequences (AC), environmental concerns (EC) and perceived consumer effectiveness (PCE) [26]. Personal norms are significant predictors of pro-environmental behavior [27,28]. However, the evidence for whether personal norms trigger pro-environmental behavior is mixed [29,30]. While some studies do show a positive effect, others do not, which is referred to as the personal norms pro-environmental behavior gap [31]. So, to resolve this inconclusiveness of this relationship in the literature, it becomes necessary to consider another variable as a moderator to strengthen the relationship of personal norms and pro-environmental behavior. In support of this view, we employ social media as a moderating variable to examine the predictive power of the personal norm and pro-environmental behavior relationship. Empirical evidence points to the fact that social media has become part and parcel of pro-environmental behavior [30]. Social media, the internet and mobile technology have become essential and useful communication instruments in connecting people all over the world. Previous studies have studied the role of social media for several social reasons, like green purchasing, environmentally-friendly actions, recycling intention or reduction in fast food consumption [32,33]. However, the current understanding of biogas technology adoption from a social media perspective is still lacking in the literature, especially in the context of Pakistan, where the Populationof internet users is approximately 71 million, from which 32 million have a Facebook account. Hence, this study is an attempt to bridge this gap in the literature by studying the moderating role of social media in the relationship between personal norms and biogas technology adoption.

## 2. Literature Review

This section covers the detailed review of the energy situation and potential of biogas technology in Pakistan. After analyzing the scenario, the theoretical support of NAM was taken, and hypotheses were developed based on the previous literature.

### 2.1. The Energy Situation in Pakistan

The proper availability and easy accessibility of energy are the essential elements for individual quality of lifespan, employment opportunities and economic growth in any country. The socio-economic growth is heavily based on the energy stream and is considered the backbone of an economy. Developing countries like Pakistan are suffering from severe energy problems, and this crisis further leads to adverse economic, social and environmental issues in the long term [34]. The energy consumption in Pakistan is increasing with every passing day due to the growing population and the demand for energy is rising by approximately 11–13 percent per year. Fossil fuels like coal, oil and gas are the primary energy sources in Pakistan and are an expensive source of energy with an annual expenditure of nearly seven million US dollars [35]. Pakistan generates approximately 61 percent of electricity from expensive sources such as crude oil and gas. However, the country is still facing a deficiency in electricity of about 14–18 h in rural areas and 8–10 h in urban areas since 2006 [36]. To meet its energy needs, Pakistan spends almost 1.4 billion US dollars on the import of fossil fuels, which causes adverse effects on economic growth because this fuel cost was only 530 million US dollars in 1996 [11]. The demand for energy has been estimated to rise threefold by 2050 in Pakistan, but the electricity production rate is shallow and is not even able to meet the current demand for energy in the country [37]. The neglect of sustainable energy sources, higher dependency on fossil fuels and lack of knowledge are the leading causes of energy crises in Pakistan [38]. Meanwhile, all sectors in the country are adversely hit by the energy shortfall, but the textile industry faces the most acute energy crisis; in recent years, about 50 to 60 percent of textile industries shifted their businesses to Bangladesh and China [13].

### 2.2. Biogas Potential in Pakistan

Biogas is the fourth leading energy source in the world that provides almost 14 percent of primary energy and power [2]. Biogas is one of the best alternates of fossil fuel, fuel wood, agricultural residues and animal dung, which are mostly used as energy and cooking sources in rural areas. In developing countries like Pakistan, biomass energy has a prime significance due to multiple factors such as the production of low-cost electricity, small capital investment and, most importantly, the easy availability and accessibility of its raw material [13]. Biogas technology produces hygienic and safe energy which is environmentally friendly and does not compromise human health. Pakistan is an agricultural and agro-livestock country, which has excellent potential to provide a significant quantity of animal dung [39]. In Pakistan, there are more than 170 million livestock animals, of which 72 million are cows and buffalo and their (approximately) 72 million kg of dung has an abundant potential to produce biogas energy in the country [40]. It is estimated that, probably, a medium size single cow and buffalo can produce 10 kg of dung per day [41] and 20 kg of wet animal dung can provide approximately one cubic meter of biogas. Besides, bagasse, poultry waste, slaughterhouse waste and street waste can also be used to create biogas energy in the country. It is estimated that Pakistan can produce 14.25 × 106 m^3^/day of biogas energy which is adequate to meet the needs of the 112 million people living in rural areas [42]. In Pakistan, the livestock sector is growing at a rate of four percent each year. To overcome the energy crises in the country, the Pakistan Council of Renewable Energy Technologies has installed 4016 biogas plants which have the potential to produce biogas of almost 20,454 m^3^/day [43]. Sindh was the first province in Pakistan to establish a biogas plant in 1959, and an additional 21 plants were established across the country in 1974 [18]. Currently, three significant departments, Pakistan Council for Appropriate Technologies, Pakistan Council of Renewable Energy Technologies and Pakistan Renewable Energy Society have installed 5357 biogas plants [41].

### 2.3. Norm Activation Model (NAM)

The norm activation model was proposed by Schwartz [25], and is quite popular in social psychology to study altruistic or pro-environmental behavior. According to the norm activation model, personal norms are considered as the crucial construct which drive the feelings or emotional state of personal moral responsibility to behave in a specific manner [44]. The NAM is considered a useful model to study altruistic behavior and has been mostly used to examine pro-environmental behavior such as energy preservation activities [45,46,47] and sustainable environmental grievance attitude [48,49].

This model contains three variables: the ascription of responsibility, awareness of consequences and personal norms. Ascription of responsibility (AR) is described as feelings of moral responsibility for negative or harmful effects of not performing pro-environmentally [50]. Personal norms (PR) are defined as performing moral responsibilities with specific actions, and this element of the norm activation model is useful for predicting pro-environmental behavior instantly [44]. Awareness of consequences (AC) reflects whether or not a person is aware of the adverse effects of his actions on the society or conscious about those values which are not pro-environmental [50]. From these three variables of NAM, the ascription of responsibility and awareness of consequences are antecedent variables of personal norms that can affect an individual’s behavioral intentions, plans or actions [51].

Unfortunately, NAM only covers internal factors and ignores the external factors. Many researchers improve the explanatory power of this model by adding different factors [47,52]. Scholars usually extend the norm activation model from two different perspectives, such as by introducing additional antecedents of personal norms and incorporating the external elements. In previous studies, the NAM model was extended by perceived consumer effectiveness [26,53] and environmental concern as an antecedent of the personal norm [26]. Moreover, NAM is also augmented by adding perceived behavioral control as an antecedent of the personal norm to study and understand consumer behavior [54]. Another work in this domain developed an extended norm activation by incorporating external cost variables to better explain the choice behavior of consumers [55]. Although previous studies highlighted the inclusion of external variables, inconsistencies in the relationship between personal norms and pro-environmental behavior were not incorporated by many studies [31]. In this study, we include the moderating role of social media to understand farmers’ choice behavior towards the adoption of biogas technology in Pakistan.

### 2.4. Research Framework and Hypothesis Development

Environmental concerns, perceived consumer effectiveness, the ascription of responsibility and awareness of consequences are antecedent variables of personal norms. Subsequently, personal norms can affect an individual’s behavioral intentions of adopting biogas technology. Besides, external factors (the role of social media) moderate the influence of personal norms, perceived consumer effectiveness and environmental concerns on farmers’ intentions toward the adoption of biogas technology (see Figure 1). Hence, we propose the following hypotheses:

**H1:** 
*Personal norms have a significant positive influence on farmers’ intentions toward biogas technology;*


**H2:** 
*Awareness of consequences has a significant positive influence on personal norms;*


**H3:** 
*Ascription of responsibility has a significant positive influence on personal norms.*


### 2.5. Environmental Concerns

Environmental concern indicates the consumer’s overall orientation towards environmental issues [56]. In the last few decades, ecological changes have occurred quite dramatically, and many researchers have turned their focus to environmental issues, specifically eco-friendly behavior or actions [57]. Environmental concern is one of the most crucial elements in shaping the personal norms of consumers toward environmentally friendly activities [58]. Environmental concerns are based on the consumer’s level of understanding of their ecosystem and consumers who have more knowledge about environmental issues exert more efforts to solve the ecological problems compared to those who have less knowledge about environmental issues [59]. Environmental concerns play a vital role in a consumer’s decision to adopt environmentally friendly behavior because those people who show more significant concerns about ecological problems are more willing to accept and promote renewable energy sources for the sake of environmental protection and also engage themselves in eco-friendly activities [26,60]. Therefore, we hypothesize that:

**H4:** 
*Environmental concerns positively affect consumers’ personal norms.*


### 2.6. Perceived Consumer Effectiveness (PCE)

The concept of perceived consumer effectiveness (PCE) is referred to as the psychological phenomenon in which consumers show a positive attitude towards sustainable and renewable energy sources to preserve the environment [61]. PCE is defined as how consumers justify their behavior and have faith that their behavior and actions are beneficial for the environment [45]. In the current study, perceived consumer effectiveness (PCE) is defined as the general public’s beliefs that they can contribute and assist in overcoming the negative impacts of energy consumption on the environment by adopting renewable energy sources such as biogas technology.

In previous studies, most of the researchers proposed that PCE affects the attitudinal aspects of consumers, like their attitude and subjective norms [59,62]. Consumers who are more concerned about environmental issues are more willing to work for ecological sustainability and develop a favourable attitude towards the environment [63]. Perceived consumer effectiveness can activate the personal norms of consumers. In other words, when consumers believe that their behaviors are environmentally friendly, they are probably more motivated to reduce environmental contamination.

PCE is one of the most significant and leading elements in explaining pro-environmental consumer behavior and also motivates consumers to show a positive attitude towards the adoption of sustainable options of energy production [63]. Individuals can feel pleasure when protecting the environment through their actions and feel guilty when environmental deterioration occurs through their actions. This feeling of guilt leads to the development of positive personal norms. Hence, we hypothesize that:

**H5:** 
*Perceived consumer effectiveness positively affects consumers’ personal norms.*


### 2.7. Moderating Role of Social Media

Since it was found from the literature that there are contradictions in the relationship between personal norms and pro-environmental behavior, which implies the need for external factors in this relationship [31]. Because it was found that pro-environmental behavior is not just influenced by the consumer’s personal obligation towards environment but also some other factors, the study includes social media as moderating variable to better understand and strengthen the relationship.

Media channels play a crucial role in providing sufficient and accurate information to educate society about their mutual environmental concerns [64]. Social media is becoming an essential instrument of communication, expressing interest and gaining information about what is going on. It connects people all over the globe, enabling them to understand what is going and where. This proliferation of social media not only influences the individual’s interest but also the consumer’s behavior [65]. The social causes, or justification, of green consumption behavior suggested by social groups is influencing others to behave similarly. The general public can easily see the results of green behavior through social media, motivating other people to engage themselves in pro-environmental activities.

Moreover, it is found that social media can improve comparison psychology and self-efficacy and, ultimately, these attributes contribute to promoting pro-environmental behaviors [66,67,68]. It was found that media channels have a direct influence on consumers’ attitudes and behaviors toward different kinds of environmental issues such as greenhouse gasses, energy crises and environmental deterioration [69]. Even though digital media is becoming a very popular and impressive tool for direct actions or movements, its success in enacting environmental changes or environmental protection actions or policies is still under investigation [70,71,72]. Recently, a study found the moderating role of electronic word of mouth (eWOM) in pro-environmental behavior [31]. So, based on the above discussion it can be hypothesized that:

**H6:** 
*Social media positively moderates the relationship between personal norms and farmers’ intentions toward biogas technology.*


## 3. Methodology

Pakistan is an agricultural country, with most of its agricultural land concentrated in Punjab. Sahiwal is the 14th biggest city in Punjab and the 21st biggest city in Pakistan. The Sahiwal division is one of the famous divisions in the province of Punjab due to several reasons such as its fertile agricultural land, peaceful natural environment and animal husbandry. We select the Sahiwal division for this study because it has the largest number of livestock and a favorable temperature, which are prerequisites for the production of domestic biogas digesters. The Sahiwal Division was divided into three districts: Sahiwal District, Okara District and Pakpattan District. To collect data, non-probability (purposive) sampling was employed in this research as it provides a representative sample when it is problematic to access the complete sampling frame. Moreover, purposive sampling is appropriate for theoretical generalization when it is challenging to access the whole population [73]. Animal ownership (number of animals) and landholdings (cultivated land area) are two crucial factors in the adoption of biogas technology. Hence, respondents were selected through purposive sampling in this study, and small farms with a minimum of five animals (cows and buffalo) and a minimum of 12.5 acres of cultivated land were chosen, because cultivated land is an essential factor in the adoption of biogas technology. In Pakistan, individuals are considered small farmers if they have a minimum of 12.5 acres of land in Punjab. Generally, to run a small biogas plant, a minimum of four animals (cows and buffalo) are required, producing 40 kg of animal dung/per day. Therefore, the number of livestock animals is considered as an essential factor for the adoption of biogas technology [74].

### 3.1. Sample and Procedure

Hair et al. [75] proposed a rule of thumb for sample size as a one to five ratio; hence, the questionnaire used in this study comprises 31 measurement objects which required a minimum collection of 155 (31 × 5) functional questionnaires. In this regard, a total number of 310 questionnaires were distributed, of which 207 were returned and 191 were usable for the analysis, and the response rate was 61.6 percent. The household head was the respondent in this study from whom the data was collected regarding financial attributes and how convenient it is to use biogas technology. The demographic profile of the respondents is given in Table 1.

### 3.2. Measurement Items

Items of this research have been adopted from different studies. All of the items used in this study have been measured using the five-point Likert scale, in which one symbolizes strongly disagree, and five expresses strongly agree. Items based on awareness of consequences and ascription of responsibility were constructed based on Zang et al. [76]. Items regarding personal norms were adopted from Smith and McSweeney [25,77]. Items related to environmental concern were adopted from Wang et al. [78]. Items of perceived consumer effectiveness were constructed based on Kim and Choi [56]. Items of social media were adopted by Sujata et al. [79,80]. All items adopted from the previous studies were first translated into the local language, i.e., Urdu, by two professional translators. After that, these statements were again translated into English by another translator, as per the suggestion of Brislin [81]. No significant differences were found, but some grammatical issues arose, which were resolved. After that, questionnaires were distributed for data collection in the local language. The items of the questionnaires are attached in Appendix A.

## 4. Data Analysis using Structural Equation Modeling (PLS-SEM)

Structural equation modelling (SEM) is more beneficial for statistical analysis in terms of accuracy, efficiency and convenience compared to other traditional statistical analysis methods [82,83]. SEM is a second-generation technique that covers the issues of first-generation analysis tools. It is a multivariate analysis method that can help to analyse numerous variables at the same time. Its ability to deal with complex and multiple relationships simultaneously is making it popular in business research day by day [83]. SEM has two well-known techniques: covariance-based SEM (CB-SBM) and variance-based SEM (VB-SBM) or partial least square (PLS)-SEM [84]. To select an appropriate statistical method is most important for social science researchers because the inappropriate selection of analytical methods can cause inaccurate conclusions [85]. Because data tend to have the problem of normality in social science studies, partial least square (PLS) is typically used and preferred over CB-SEM, as it addresses the issues of normality [86].

PLS-SEM is a two-stage analysis approach that includes measurement results in two steps; the first step is a measurement model, and the second step is a structural model [87]. The assessment of the measurement model includes the assessment of the inner model or reliability and validity tests. In contrast, the assessment of the structural model consists of the assessment of the outer model or hypotheses/relationships testing. This study used PLS 3.0 software for analysis.

### 4.1. Measurement Model

In this study, reliability and validity tests were conducted for all the given constructs for the measurement model evaluation. The valuation of the measurement model was based on reliability tests (internal consistency reliability and item reliability) and validity tests (discriminant validity and convergent validity) [88]. Item reliability was measured through outer loading, internal consistency reliability was measured through composite reliability (CR), and convergent validity was measured through the average variance extracted (AVE). The loadings of all items are well above the threshold value of 0.5 [89], and values are presented in Table 2. The value of composite reliability for each paradigm exceeds the cut-off point of 0.7, and AVE surpasses the suggested value of 0.5, which shows that the measurements are reliable [90]. The results of the study indicate that all the values of AVE are between 0.518 (intention to adopt Biogas technology) and 0.648 (social media), all the values of CR are between 0.809 (intention to adopt Biogas technology) and 0.851 (perceived consumer effectiveness), and all additional loadings are between values of 0.5 and 0.9. In this measurement model, all the values verify the validity and reliability. All the values are given below in Table 2 and Figure 2 and Figure 3.

For discriminant validity, the Heterotrait–Monotrait ratio of correlations (HTMT) criterion is considered more appropriate due to criticism of the Fornell–Larcker criteria by different researchers [83]. Discriminant validity is confirmed if the value is less than 0.85 [91] or 0.90 [92]. Table 3 shows that all values are less than 0.90.

### 4.2. Structural Model

After the evaluation of the measurement model, the structural model was evaluated, which checked the relationship between endogenous and exogenous variables. The assessment of the structural model is based on t values, path coefficient (β values), effect size (f^2^), predictive relevance (Q^2^) and coefficient of determination (R^2^). A 5000 resample bootstrapping procedure with 5% significance level (one-tailed) was used to test significance of the hypotheses. Results show that all six hypotheses are accepted. Personal norm (β = 0.329, t = 7.119 > 1.64, *p* < 0.05), awareness of consequences (β = 0.186, t = 4.102 > 1.64, *p* < 0.05), ascription of responsibility (β = 0.311, t = 6.940 > 1.64, *p* < 0.05), environmental concerns (β = 0.126, t = 2.795 > 1.64, *p* < 0.05) and perceived consumer effectiveness (β = 0.162, t = 4.015 > 1.64, *p* < 0.05) have a positive significant impact on the adoption intentions of biogas technology. Moreover, social media (β = 0.084, t = 3.173 > 1.64, *p* < 0.05) moderates the relationship between personal norms and intentions toward the adoption of biogas. The R^2^ value for biogas adoption is 0.468, which shows that model has substantial explanatory power for the adoption of biogas technology in Pakistan.

However, it is not an appropriate and effective approach to support a model only based on the value of R^2^ [75]. Thus, the best way is to measure the predictive relevance Q^2^ of the model. If the value of Q^2^ is higher than zero, then latent exogenous paradigms have great predictive relevance [93]. The results show that the value of Q^2^ is 0.237 for farmers’ adoption intentions, which suggests that the model has significant predictive relevance. The values of f^2^ are, 0.02, 0.15 and 0.35, which indicate small, medium and large effects, respectively [94]. Therefore, the value of f^2^ postulates that effect size varies from medium to large (see Table 4).

### 4.3. Moderating Role of Social Media

The moderating effect of social media is studied by the interaction effect on the adoption intentions. The findings of this study demonstrate that social media significantly (β = 0.084, t = 3.173 > 1.64, *p* < 0.05) moderates the relationship between personal norm and the intentions of adopting biogas technology (see Figure 4). This moderation has changed and the value of R^2^ has increased from 0.468 to 0.478. This means that after the addition of social media, the explanation power of the model is increased. Although the change is not very large, it plays an important role in examining the relations between personal norms and intentions of adopting biogas technology.

## 5. Discussion

The current situation of industry, modernization and high growth in population is leading to worse environmental issues and acute energy crises in Pakistan. Most of the energy is produced from fossil fuels, like crude oil and gas, which emits harmful gases. Rural areas heavily rely on fossil fuels or fuel wood and produce hazardous gas, causing respiratory diseases, particularly for women and children because fossil fuels emit an extensive amount of smoke [41]. To protect the environment, it is imperative to explore and adopt the best alternatives and renewable energy sources. Biogas is environmentally friendly because it is a smoke-free gas, has zero contribution to global warming, and is suitable for women’s health while cooking. Biogas technology is very cost-effective and does not require a large expense. Other neighbouring countries of Pakistan, like China and India, are utilizing biogas technology at a very economical cost [95]. Biogas plants are based on animal dung; thus, to run biogas plants, enough animals are required. Pakistan is an agricultural country with more than 170 million livestock animals. Normally, to run a small-scale biogas plant, 40 kg/day of animal dung is sufficient [96].

In this study, we explored the significance of altruistic value in determining farmers’ intentions toward biogas technology in Pakistan. The findings of this study show that altruistic beliefs (environmental issues) have a significant influence on farmers’ intentions toward biogas technology. Based on the norm activation model, this study explores the influencing elements for Pakistani people toward biogas technology adoption with the moderating role of social media. The extended version of NAM has been applied in this research, which can better explain the intentions toward biogas and provide the theoretical foundation for this study. Environmental concerns, awareness of consequences, the ascription of responsibility, personal norms, perceived consumer effectiveness and the role of social media are six variables that are used to explain farmers’ intentions toward biogas technology. From the specific analysis of these six factors, we found that environmental concerns, awareness of consequences, the ascription of responsibility and personal norms has a significant positive impact on farmers’ intentions toward biogas adoption. Personal norms have the most substantial influence and are the most crucial factor, and a considerable predictor of eco-friendly behavior, and these findings are in line with the results of Schwartz [25]. The results of this study support the hypothesis H1 that personal norms have a significant positive influence on farmers’ intentions toward biogas technology. Many previous studies have validated the role of personal norms in predicting the intentions toward adoption behavior [48,97,98]. Hence, these findings are helpful to determine a farmer’s possible intentions of adopting and using biogas technology.

We also found that both awareness of consequences and ascription of responsibility have a significant positive influence on personal norms and this is consistent with the statement that Personal Norm (PN) in the norm activation model can be triggered by awareness of consequences and ascription of responsibility as moderating factors [51]. The findings of this research show that the two main components of NAM, awareness of consequences and ascription of responsibilities, have a positive effect on the personal norms associated with the use of biogas technology. Hence, these findings support our hypotheses H2 and H3. These findings are consistent with the previous results that personal norms can be triggered by identifying the responsibility and consequences of the particular behavior [98]. Simply, we may say that awareness of consequences and ascription of responsibility plays a vital role in developing and strengthening personal norms. Consequently, it is more likely that farmers will be motivated to use biogas technology because of the awareness that the overuse of non-renewable energy sources can be harmful to their health and the environment. Awareness of consequences is considered an early phase in the direction of responsible behavior [25].

When individuals are more concerned about environmental issues, it is easy to promote renewable energy sources, and this statement also correlates with the previous findings that environmental concerns profoundly affect personal norms [99]. Awareness of the consequences of ecological issues plays an essential role in making and performing pro-environmental behavior [27]. When individuals have more knowledge and awareness about the environmental consequences, they are be more thoughtful and sensitive towards others and about how they consider and evaluate ecological concerns [97]. The findings of this study also support the hypothesis H4 that environmental concerns positively influence personal norms. It can be concluded that when farmers are more informed about the outcomes of biogas plants, they will have more favorable intentions toward biogas technology.

Moreover, the findings of this study support the hypothesis H5 that perceived consumer effectiveness is a cogent antecedent which positively affects consumers’ personal norms. These findings are aligned with the previous results that perceived consumer effectiveness has a significant positive influence on adoption intentions [63], as perceived consumer effectiveness is one of the most prominent elements in the explanation of pro-environmental behaviors that can positively affect behavioral intentions [56,59,100].

Social media is found to be a strong predictor that has a significantly positive influence on behavioral intentions and supports the hypothesis H6 that social media positively moderates the relationship between personal norms and farmers’ intentions toward biogas technology. Different social media platforms play a vital role in educating people about environmental issues by sharing images and messages. Extant research also found that social media influences individual behavioral intentions [101,102,103]. Posting videos, text, pictures and links associated with social, environmental and health issues increases the possibility that one will engage in more or less pro-environmental activities. Analysis of the study shows that using social media is a significant mechanism for expressing opinions, ideas and joining national causes. Hence, the findings of this study concurred with previous findings that social media sites are beneficial in developing favourable attitudes in terms of environmental protection activities [79]. Besides, the results of this study support the arguments that technology facilitates the provision of alternate solutions for environmental sustainability [104]. Therefore, we recommend that social media is a valuable tool to educate people with proper and meaningful information about environmental degradation and encourage them to promote renewable energy sources. The popularity and use of social media have given opportunities to government agencies and NGOs to promote renewable energy sources. Similarly, relevant specialists and opinion leaders have the responsibility of designing their content properly and more attentively to interact with ordinary people on social media platforms in order to promote renewable energy sources such as biogas technology. This is because, in Pakistan, approximately 72 million people use the internet and social media websites.

### 5.1. Practical Implications

The demand for electricity is increasing with every passing day and Pakistan is pressurized to import both conventional or unconventional sources of energy to meet the electricity demand in the country. On average, the energy shortfall in the country is 5000 MW, and this shortfall touches up to 7000 MW in June and July when the demand for energy is at its peak. However, animal waste in the country has the potential to produce almost 4761–5554 MW electricity [11]. The outcomes of this study have significant practical implications for economic growth as well as for the wellbeing of rural farmers in term of energy generation and biofertilizers. Biofertilizers can improve the productivity or efficiency of the soil. This research provides ideas and alternate solutions to cope with energy crises via renewable energy sources such as biogas technology, with the consideration of personal norms, awareness of consequences and ascription of responsibility, environmental concerns and perceived consumer effectiveness to enhance the intentions in the long run. Personal norms highly influence individuals’ enthusiasm toward the adoption of renewable energy sources. Government agencies are beginning to pay more attention to personal norms to promote renewable energy sources such as biogas technology. The government of Pakistan has set the target to increase the share of renewable energy sources from less than one to five percent by 2030. In this regard, biogas technology has substantial potential to achieve this growth rate. To achieve such targets, education and publicizing the environmental benefits of biogas, while also highlighting the moral obligations is essential because moral obligations motivate individuals to adopt biogas technology.

It is noted that rural farmers are not considering biogas technology seriously due to their lack of knowledge and awareness. Promotion of biogas adoption through media channels can be helpful, along with educating the female members of the household. The outcomes of the moderating role of social media are also useful for policymakers and practitioners in promoting biogas technology, as there is a vast segment of the population using the internet and social media websites. Therefore, government agencies can use social media and other media platforms to convey the harmful consequences of using non-renewable energy sources such as global warming, environmental degradation and risks associated with the overuse of fossil fuels, coal or liquefied petroleum gas (LPG). As a result, people will show more favourable behavior toward renewable energy sources to resolve the energy crises and environmental issues.

Furthermore, rural farmers have some reservations in the adoption of biogas technology, such as initial investment. To overcome such impediments, the government shall provide subsidies, interest-free loans and arrange proper training programs to assist in the installation and function of biogas plants. Likewise, the private sector and NGOs can also take initiatives such as offering sponsorships and provision of small and easy loans for the installation of biogas plants, because if the installation cost or risks associated with biogas are high, the adoption intentions of this technology will slow down [105]. Consequently, in Pakistan, biogas is produced through animal dung at a limited or local level, though it can be taken up to a large or commercial scale if the government pays proper attention and formulates policies and strategies to promote this technology.

The empirical outcomes of this research are appropriate and applicable not only in the context of Pakistan but can also be helpful for other countries because the norm activation model (NAM) has universal acceptability for pro-environmental behaviors and energy issues that exist around the globe.

### 5.2. Limitations and Future Research Directions

Biogas technology will be the most crucial substitute for energy in Pakistan in the upcoming years. Some domains still need work or the attention of researchers in the development of biogas technology in Pakistan. The study is only limited to the intention/adoption of biogas technology. Still, there is always a gap in actual behavior and the intention of consumers, so future studies can investigate the actual buying behavior of the farmers. Secondly, this integrative model could only describe 47.8 percent of the variance of farmers’ intentions to use biogas technology. Therefore, it is suggested that some additional variables from other relevant models, such as the trans-theoretical model (TTM), innovation diffusion theory (IDT) and technology acceptance model (TAM), can also influence farmers’ intentions. Hence, these models can also be employed along with the NAM to improve the explanatory power of the proposed model. Furthermore, in this study, we only use the role of social media, but other media channels like broadcast, print and electronic media are also used to familiarize rural society with biogas technology adoption; thus, we also suggest extending the norm activation model with such factors. Future studies may also be conducted on the safe and nontoxic storage of biogas energy that is produced locally.

## 6. Conclusions

This study employs an extended version of the norm activation model because the model is suitable for describing the individual’s adoption intentions in pro-environmental issues. To extend the model, we integrate internal variables with external variables. With the help of NAM, we studied the relationship between personal norms and intentions toward the adoption of biogas technology in Pakistan. We also investigated the moderation effect of social media between personal norms and adoption intentions and also inspected the antecedents of personal norms, such as ascription of responsibilities, awareness of consequences, perceived consumer effectiveness and environmental concerns. The empirical results of this study enhance our understanding of the practical and theoretical implications of farmers’ intentions to use biogas technology in Pakistan.

Furthermore, we investigated the economic, environmental and health benefits of adopting biogas technology as a renewable source of energy in Pakistan. Out of all renewable sources of electricity, biogas is considered one of the most suitable alternatives that is not only economical, but also an environmentally friendly source of energy. Pakistan is an agricultural country and has abundant resources of raw materials to run biogas plants. In Pakistan, the agriculture and livestock sectors have enormous potential and prospects, as they produce animal waste in massive quantities. Currently, there is no significant use of animal dung, and this waste causes environmental hazards because animal waste is thrown in open land areas, which can cause serious health issues. Hence, this technology is not only beneficial for farmers and households, but also beneficial for economic growth and environmental preservation. Thus, the present study highlights that biogas technology is one of the most essential and suitable alternatives for energy production in Pakistan.

## Figures and Tables

**Figure 1 ijerph-17-02311-f001:**
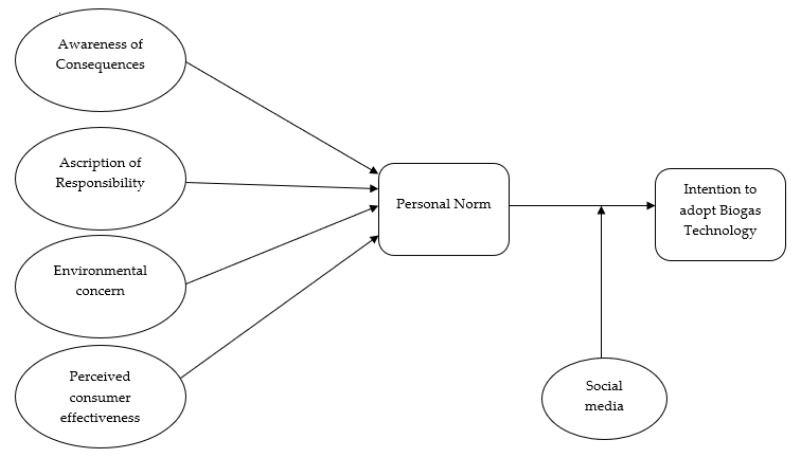
Conceptual framework.

**Figure 2 ijerph-17-02311-f002:**
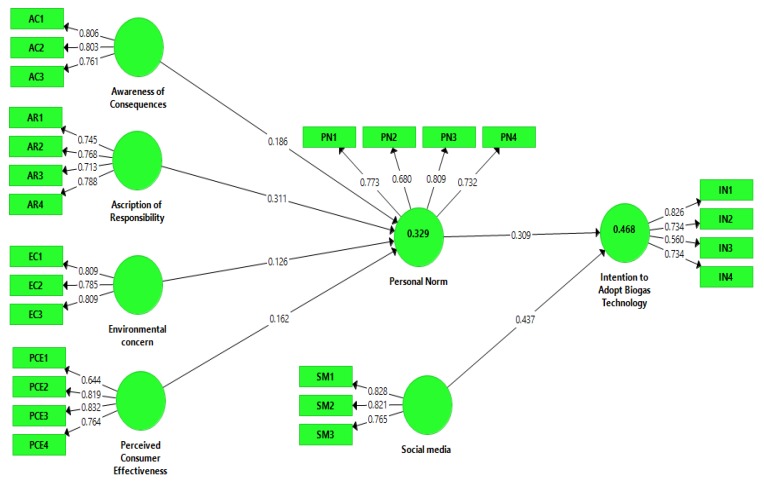
Loadings.

**Figure 3 ijerph-17-02311-f003:**
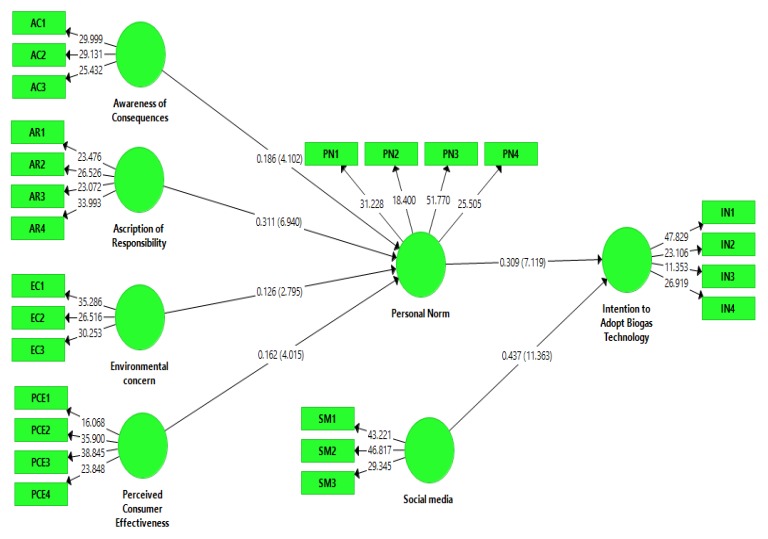
Structural model evaluation (bootstrap).

**Figure 4 ijerph-17-02311-f004:**
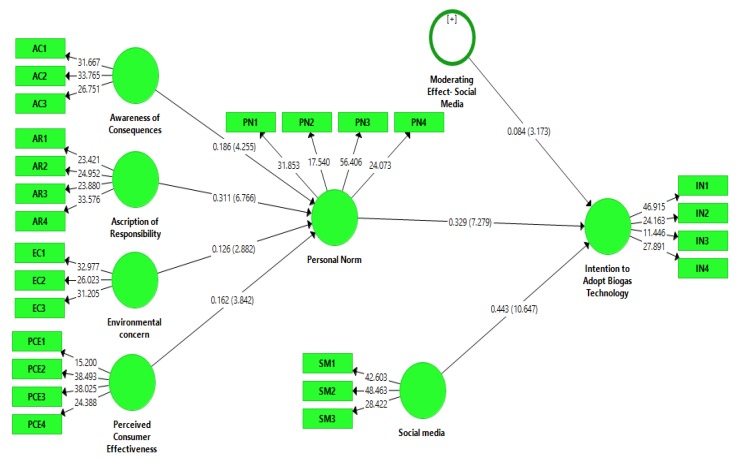
Moderating effect of social media.

**Table 1 ijerph-17-02311-t001:** Demographic profile.

Variables	Characteristics	Frequency	Percentage
Gender	Male	117	61.26
	Female	74	38.74
Age	Less than 20	26	13.61
	21–30	58	30.37
	31–45	42	21.99
	46–55	35	18.32
	56–65	19	9.95
	65 and above	11	5.76
Income	Less than 50,000	35	18.32
	50,001–75,000	62	32.46
	75,001–100,000	46	24.08
	100,001–125,000	33	17.28
	125,001–and over	15	7.86
Head of Cattle	5–10	38	19.90
	11–15	48	25.13
	16–20	65	34.03
	21–25	18	9.42
	26– and above	22	11.52
Cultivate Land area	12.5–20	43	22.52
	21–30	82	42.93
	31–40	40	20.94
	41– and above	26	13.61

**Table 2 ijerph-17-02311-t002:** Measurement model assessment.

Constructs	Item	Loading	CR	AVE
Personal norm	PN1	0.732	0.836	0.562
	PN2	0.680		
	PN3	0.809		
	PN4	0.732		
Awareness of consequences	AC1	O.806	0.833	0.625
	AC2	0.803		
	AC3	0.761		
Ascription of responsibility	AR1	0.745	0.840	0.569
	AR2	0.768		
	AR3	0.713		
	AR4	0.788		
Environmental concern	EC1	0.809	0.843	0.642
	EC2	0.785		
	EC3	0.809		
Perceived consumer effectiveness	PCE1	0.644	0.851	0.590
	PCE2	0.819		
	PCE3	0.832		
	PCE4	0.764		
Social media	SM1	0.828	0.847	0.648
	SM2	0.821		
	SM3	0.765		
Intention to adopt Biogas Technology	IN1	0.826	0.809	0.518
	IN2	0.734		
	IN3	0.560		
	IN4	0.734		

**Table 3 ijerph-17-02311-t003:** Discriminant validity Heterotrait–Monotrait (HTMT) table.

Constructs	AR	AC	EC	IN	PCE	PN	SM
AR							
AC	0.481						
EC	0.455	0.586					
IN	0.440	0.663	0.543				
PCE	0.373	0.565	0.445	0.521			
PN	0.614	0.557	0.478	0.812	0.488		
SM	0.585	0.548	0.431	0.869	0.575	0.889	

**Table 4 ijerph-17-02311-t004:** Structural model results (hypotheses testing).

Hypothesis	Relationship	Path Coefficient	Std. Error	t Value	*p*-Value	Supported	R2	Q2	f2
H1	PN→IN	0.309	0.043	7.119	0.000	Yes	0.468	0.237	0.099
H2	AC→PN	0.186	0.046	4.102	0.000	Yes		0.179	0.037
H3	AR→PN	0.311	0.045	6.940	0.000	Yes			0.118
H4	EC→PN	0.126	0.043	2.795	0.000	Yes			0.018
H5	PCE→PN	0.162	0.040	4.015	0.000	Yes			0.031
H6	Moderating effect SM→IN	0.084	0.024	3.173	0.001	Yes	0.478		0.019

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
