# Peer review of "Determining the Influencing Factors of Biogas Technology Adoption Intention in Pakistan: The Moderating Role of Social Media"

_ijerph, 2020, doi:10.3390/ijerph17072311_

Round 1

Reviewer 1 Report

The paper explores through the use of a structural equation model the influencing factors affecting farmers intention to adopt biogas technologies in Pakistan. The paper is quite readable even if improvement is possible.

The paper does not specify if alternative models have been tested in order to assess the superiority of the theoretical model proposed. E.g. in the model presented social media affects the farmers intention to adopt biogas technology, but could also have direct effect on farmers personnel norms or influence perceived consumers effectiveness, that is to say, the role of social media in shaping farmers perceptions in relation to environmental issues and the impact of farmers perceived consumers pro-environment behaviours and choices directly affecting the farmers.

The questionnaire (translated in English) should be added in an annex 

A check of the text should be carried out to improve readability and ensure the consistency of data presented, e.g. number of internet users (71 or 72 million), per day production of dung by cows and buffalos (lines 160-162), numeric expressions (line 169, 14.25 × 106 m3/day)

Author Response

Reviewer corrections submitted

Reviewer 2 Report

Please review my comments in the attached file for minor corrections.

Author Response

Reviewer 2 comments submitted 
